## PERSPECTIVE

# Intracellular thermometry uncovers spontaneous thermogenesis and associated thermal signaling

Kohki Okabe [1,2✉] & Seiichi Uchiyama [1✉]

Conventional thermal biology has elucidated the physiological function of temperature homeostasis through spontaneous thermogenesis and responses to variations in environmental temperature in organisms. In addition to research on individual physiological phenomena, the molecular mechanisms of fever and physiological events such as temperature-dependent sex determination have been intensively addressed. Thermosensitive biomacromolecules such as heat shock proteins (HSPs) and transient receptor potential (TRP) channels were systematically identified, and their sophisticated functions were clarified. Complementarily, recent progress in intracellular thermometry has opened new research fields in thermal biology. High-resolution intracellular temperature mapping has uncovered thermogenic organelles, and the thermogenic functions of brown adipocytes were ascertained by the combination of intracellular thermometry and classic molecular biology. In addition, intracellular thermometry has introduced a new concept, "thermal signaling", in which temperature variation within biological cells acts as a signal in a cascade of intriguing biological events.

Temperature is one of the most influential physical parameters for human beings. During the cold times of the year, people enjoy traveling to tropical islands under the warmth they miss in the winter months. Seasonally, we sweat in summer and shiver in winter to maintain a constant body temperature. When we feel ill, one of the first things that people do is reach for a thermometer to see if they are sick. Regardless of recognition, our daily lives are deeply impacted by temperature and its variation. In biology, the temperature has a profound effect on the internal activities of living things, and thus, the relationship between temperature and organisms has long been a crucial target of scientific understanding (Box 1). In both commonplace and academic cases, thermal detection devices such as alcohol-filled thermometers, thermistors, and infrared radiation detectors are important instruments to accurately inform us of an object's temperature. Molecular thermometers are capable of downsizing the target of temperature measurement to the micro-meter scale or smaller, which includes biological cells. In this Perspective, we summarize how temperature has been investigated in the history of biological studies. An established field, called thermal biology, correlates temperature variation with the diverse functions of individual organisms and elucidates their molecular mechanisms. Most noteworthy; in recent years new developments in thermometry have allowed for temperature measurements to be performed at the single-cell level, which has revealed that in addition to the thermogenic properties of specific cells and organelles, temperature variation is a driving force of biological events.

### Temperature of individuals

The body temperature of endotherms is kept higher than that of the environment by spontaneous heat generation, and it fluctuates greatly in association with physiological activity[1].

[1] Graduate School of Pharmaceutical Sciences, The University of Tokyo, Tokyo, Japan. [2] JST, PRESTO, Saitama, Japan. ✉email: okabe@mol.f.u-tokyo.ac.jp; seiichi@g.ecc.u-tokyo.ac.jp

**Box 1 | Keywords in biological studies that correlate temperature and physiological events. These indices are explained by representative literature**

Body

Cell

Spontaneous thermogenesis

**Body temperature**
Circadian rhythms
Menstrual cycle
Vital sign

**Regulation mechanism**
Neural circuits
Fever due to infection and stress

**Cellular thermometry**
Fluororescent molecular thermometers

**Thermogenic organelles**
Nucleus and mitochondria

**Thermogenic brown adipocyte**

Response to environmental temperature

**Adaptation**
Homeostasis
Hibernation
Plants flowering

**Utilization**
Sex determination

**Heat shock response**
Heat shock proteins
Stress granule

**Thermosensitive TRP channels**

Humans have considerable variability in body temperature in relation to circadian rhythms, including ultradian and infradian rhythms, and moreover in relation to race, age, voluntary exercise, and disease[2]. In addition, women have another circadian rhythm of basal body temperature due to the ovulation cycle[3]. Although the menstrual cycle-dependent variability in basal body temperature is affected by aging, the effect of seasonal variation is small, and basal body temperature is being proposed as a noninvasive diagnostic indicator of ovulation[4].

Body temperature is one of the most basic vital signs in clinical diagnosis and routine health care[5,6]. To measure body temperature variability with high accuracy, actual and predictive measurements using instruments or mathematics, either invasive or noninvasive, are used[5]. In recent years, new methods (wearable temperature sensors[7] and information technology-based real-time and long-term measurements[5]) and applications (life-critical decision making and mass screening of diseases[6]) of body temperature measurements have been proposed.

The regulation of body temperature is one of the most important functions of the nervous system, and the molecular mechanisms of temperature sensing in the periphery and the neural circuits that transmit temperature information to the brain, as well as the central circuits for maintaining body temperature homeostasis, have been determined[8]. In addition to the inflammatory response induced by pathogen infection[9], fever is also caused by social stress[10], and these neural circuits have been identified. Among the thermoregulatory mechanisms in the brain, molecular entities and neural circuits in temperature sensing are still unclear[8].

The adaptation of body temperature in response to environmental temperature variation is an essential function of life for endotherms and is remarkably diverse. Humans spontaneously generate heat and maintain homeostasis even in cold environments[1], and the disruption of homeostasis can lead to serious life threats. During hibernation, even in cold environments where metabolic activity is markedly suppressed, black bears maintain a body temperature of 30–36 °C by performing regular muscular exercises[11] without waking up from sleep[12]. Sea turtles living in water maintain a deep body temperature 0.7–1.7 °C higher than that of the water due to the size-dependent physical effects of high heat production rates[13].

Plants have evolved excellent plasticity to adapt to their surrounding temperature environment[14]. For example, the flowers and inflorescences of primitive seed plants regulate their heat production rate during flowering to remain at a much higher temperature than their surroundings[15]; *Symplocarpus renifolius* is able to generate a large amount of heat during the female flowering stage owing to the enhanced activity of mitochondria in the reproductive organs[16]. Unlike those of many previously described organisms, the mechanisms by which plants sense changes in ambient temperature and produce heat have not yet been elucidated[14,15].

On the other hand, there are some biological phenomena in which environmental temperature is utilized to determine fate. For example, in lizards and turtles, it has been discovered that sex is determined by environmental temperature; the effect of incubation temperature on the reproductive success of males is different from the effect on females[17]. The corresponding genes responsible for this temperature-dependent phenomenon have been identified[18].

## Temperature sensing at the cellular level

When living organisms are subjected to an environmental temperature change, temperature-sensitive proteins initially respond to it. One of the typical thermal responses at the cellular level is

the heat shock response[19]. When an organism is exposed to an excessive temperature of 5–20 °C above the normal growth temperature, the expression of heat shock proteins (HSPs) is induced in the cell. HSPs protect cells from heat damage by preventing the aggregation of functional proteins through modulation of the transcription of related genes[19]. The cellular response to thermal stimuli also includes fast translational regulation by changes in the state of translating mRNAs, which are mediated by stress granule formation[20]. Thermosensitive transient receptor potential (TRP) channels also play important roles in temperature sensing at cell membranes exposed to large temperature fluctuations. A variety of TRP family proteins are activated at each specific temperature and implicated in temperature-related physiological functions[21]. Another interesting temperature-responsive protein that has been discovered is the splicing factor, which responds to temperature fluctuations[22]. This protein can detect a temperature change of 1 °C for use as an input factor in regulating gene expression. Intracellular thermo-responsive molecules such as a heat shock promoter[23] and/or a temperature-sensitive (ts) mutant protein[24] would allow for functional analyses of any given protein via spatiotemporally controlled heating within a cell.

## The benefits of intracellular thermometry

Just as we maintain our body temperature higher than the environment, so do cells spontaneously produce heat. In addition, environmental temperature greatly influences cellular temperature to provoke various temperature responses, as described in the previous section. These universal facts inevitably led us to explore the temperature inside cells. The efforts to measure intracellular temperature began with the placement of fluorescently labeled lipids[25] or a hydrophobic thermosensitive dye[26] on cell membranes. Around 2010, our group[27] and Yang et al.[28] pioneered the detection of the temperature fluctuations inside of single living cells by combining fluorescent molecular thermometers introduced inside cells with optical microscopy. These achievements have accelerated the further establishment of intracellular temperature measurement methods using various fluorescent molecular thermometers or nonfluorescent metal materials represented by thermocouples; for more details about each method, their shortcoming, and uncertainties, as well as potential improvements, readers can refer to the invaluable review articles[29–35]. Thus, we here focus on the contribution of intracellular thermometry to thermal biology, avoiding repetitive descriptions of the comprehensive characteristics of the methods themselves. Now, a temperature variation of <1 °C can be detected in a cell, and new insights into cellular temperature have been intensively uncovered. For example, non-uniform temperature distribution due to spontaneous intracellular local heat generation was recorded[28]. The organelle-specific thermogenesis was observed in the nuclear region (Fig. 1a, b)[27,36] and near mitochondria in steady-state cells[27]. The intracellular temperature is also affected by the cell cycle[27,37]. Furthermore, chemical stimuli-induced intracellular temperature elevations have been reported, in which ATP synthesis in the mitochondria is perturbed (Fig. 1c)[27,38].

## Evaluation of thermogenic activity in brown adipose tissue (BAT) cells.

One of the most active fields to which intracellular thermometry contributes is monitoring the thermogenesis of BAT cells. BAT cells are morphologically characterized by well-developed mitochondria and fat droplets and are responsible for the maintenance of steady-state body temperature and its increase in response to an external stimulus. Different research groups have tracked the intracellular temperature increase due to the

signaling cascade leading to $\beta_3$-adrenoceptor agonist-induced thermogenesis in BAT cells, with a focus on the roles of apoptosis signal-regulating kinase 1 (ASK1) (Fig. 1d)[39], norepinephrine (NE, Fig. 1e)[40], A-type natriuretic peptides (ANPs)[41], isoproterenol[42], protein kinase R (PKR)-like endoplasmic reticulum kinase (PERK)[43], pannexin-1 (Panx1) channels[44], and selenoprotein P (SeP)[45] (Fig. 1f). Furthermore, the temperature increase in BAT cells was found to depend on mitochondrial uncoupling protein 1 (UCP1), and heat production was suppressed in cells lacking the gene required for the expression of UCP1[39,43,44]. Notably, these intracellular temperature changes in BAT cells have been correlated with the phenotype of temperature maintenance and respiratory activity in individuals[39,43,44]. Moreover, the results of intracellular thermometry in BAT cells are under discussion for their association with body temperature[46] and efficient hypothermia therapy[47] in humans. Temperature measurement studies in BAT cells exhibit direct evidence that temperature elevations at the cellular level cause a temperature increase in the whole body in individuals.

## Unveiled physiological significance of intracellular temperature variations.

Another remarkable insight brought by intracellular thermometry is that the intracellular temperature is ingeniously controlled. Recently, a switch was discovered in mitochondria that transitions between energy production (ATP synthesis) and thermogenesis (Fig. 2a)[48]. Quantitative heating in *Caenorhabditis elegans* embryos calibrated by intracellular thermometry, with fluorescent nanodiamonds having relatively low chemical interference[49], promotes embryonic development in a fixed manner, indicating that embryogenesis is controlled by intracellular temperature (Fig. 2b)[50]. Studies on cell temperature measurement in brain tissues revealed that ischemic stimuli, such as traumatic accidents, increase spontaneous heat generation (Fig. 2c). This intracellular temperature increase due to enhanced neuronal activity triggers the opening of the TRPV4 channels of the cell, followed by the progression of severe brain edema[51]. This is an important example in intracellular thermometry because the temperature rise associated with cellular activity leads to another physiological response.

## Promising future of intracellular thermometry

In 2014, Baffou et al., assuming the cell as a liquid with a single composition, pointed out that the possible intracellular temperature change ($\sim 10^{-5}$ °C) calculated from the heat conduction equation established by Fourier in the 1820s deviated significantly from the early reported experimental values (0.1–1 °C)[52]. Although we still find some scientists flinching at this gap, we contend strongly that this concern is unnecessary. First, even after this discussion by Baffou et al. various intracellular temperature measurement methods have independently observed temperature changes of 0.1 °C or more, confirming the high reproducibility of the initial results. This is a common and important process by which the reliability of a novel methodology (i.e., intracellular thermometry in the present case) is experimentally established[53]. Second, a careful examination of the heat conduction equation used by Baffou et al. shows that it does not accommodate any heat biologically consumed within a cell. Cells keep entropy low, a phenomenon found exclusively in living organisms[54]. Much energy is expended to regulate the high-dimensional structure of biopolymers such as membranes, proteins, and nucleic acids and to organize controlled one-way irreversible chain reactions. In actual biological cells, many endothermic reactions occur[55], and the energy stored in this way may be dissipated through subsequent exothermic reactions. It has been suggested that the heat generated by enzymatic reactions is used for the diffusion of the

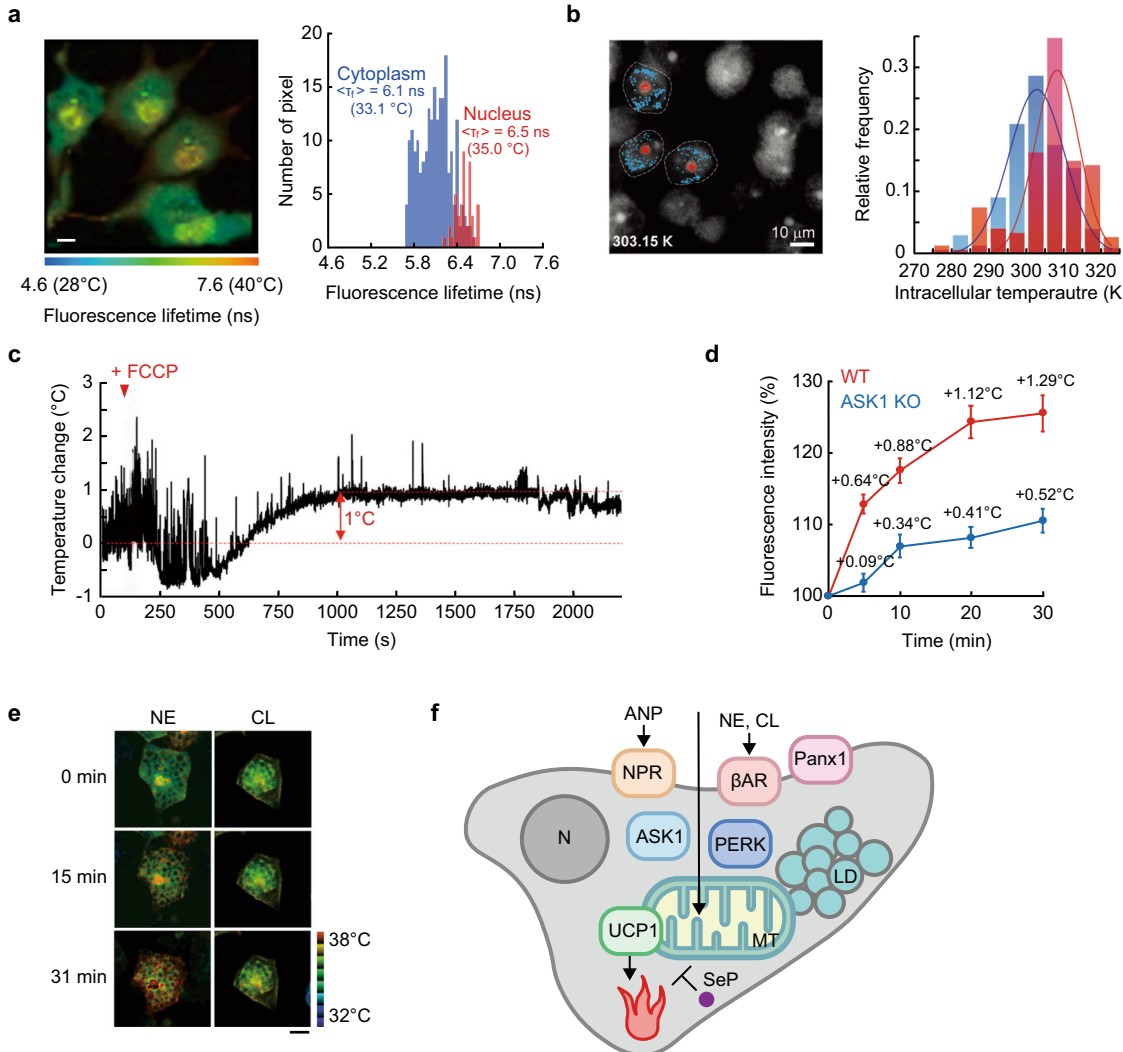

**Fig. 1 Thermogenesis revealed by cellular thermometry. a, b** Location-dependent thermogenesis in single living cells revealed by fluorescent molecular thermometers. The nucleus in COS7 cells (**a**) and nucleolus in MDA-MB-468 cells (**b**) showed higher temperatures than the surrounding cytoplasm. Scale bar, 10 µm. **c** Detection of cellular temperature changes due to mitochondrial thermogenesis upon 4-(trifluoromethoxy) phenylhydrazone (FCCP) stimulation (added at 100 s) in COS7 cells with vanadium dioxide ($VO_2$) microthermistors. **d–f** Thermogenesis in brown adipocytes upon stimulation shown by fluorescent molecular thermometers. **d** ASK1-dependent thermogenesis upon $\beta_3$ adrenoreceptor-specific agonist CL316.243 (CL) [$n = 11$ (WT), 22 (ASK1KO) cells]. **e** Intracellular temperature imaging in brown adipocytes after NE or CL treatment. Bar, 20 µm. **f** Schematic drawing of the molecular mechanism of heat production in brown adipocytes. Physiologically active molecules such as A-type natriuretic peptide (ANP) and β adrenergic agonists (NE and CL) are detected by receptors such as natriuretic peptide receptor (NPR), β adrenoreceptors (βAR), and Pannexin1 (Panx1) at the cell membrane, followed by cell signaling, including ASK1 and PERK, to the mitochondria (MT), where the uncoupling reaction by UCP1 is provoked to produce heat that can be inhibited by SeP. N and LD indicate nucleus and lipid droplets, respectively. Panels **a**, **d**, and **e** adapted from refs. [27, 39, 40], respectively, Springer Nature. Panel **b** adapted with permission from Piñol, R., et al. Real-time intracellular temperature imaging using lanthanide-bearing polymeric micelles. *Nano Lett.*, **20**, 6466–6472, copyright 2020 America Chemical Society[36]. Panel **c** adapted from Inomata, N., Inaoka, R., Okabe, K., Funatsu, T. & Ono, T. Short-term temperature change detections and frequency signals in single cultured cells using a microfabricated thermistor. *Sens. Biosensing Res.*, **27**, 100309, Copyright (2020)[38], with permission from Elsevier.

enzymes themselves[56]. At present, it is impossible to precisely quantify the processes by which heat generated within a cell is dissipated as other types of energy in addition to heat conduction. Thus, it would be an error to treat dissipation of heat as if it were solely due to intracellular heat conduction. The dissipation of energy converted from heat is no longer heat conduction, and the consequence of this conversion on the temperature change in a cell, whether greater or lesser, is an open question. Even following the application of the heat conduction equation, several research groups have reevaluated the cellular thermal conductivity using lipid bilayers[57] and live cells[58,59] to narrow the gap that Baffou et al. pointed out. These results have cast doubt on the validity of

Baffou's comment and supported our justification of utilizing intracellular thermometry in biological research.

Considering that intracellular temperature is an extremely influential physical parameter for thermodynamics, thermochemistry, and heat transfer in a cell, it seems impossible to explain intracellular temperature variation by only one scientific field. Here, we refer to the statement by the physicist Schrödinger in his lecture in 1943 in Dublin[60]:

> "What I wish to make clear …is… we must be prepared to find it [living matter] working in a manner that cannot be reduced to the ordinary laws of physics."

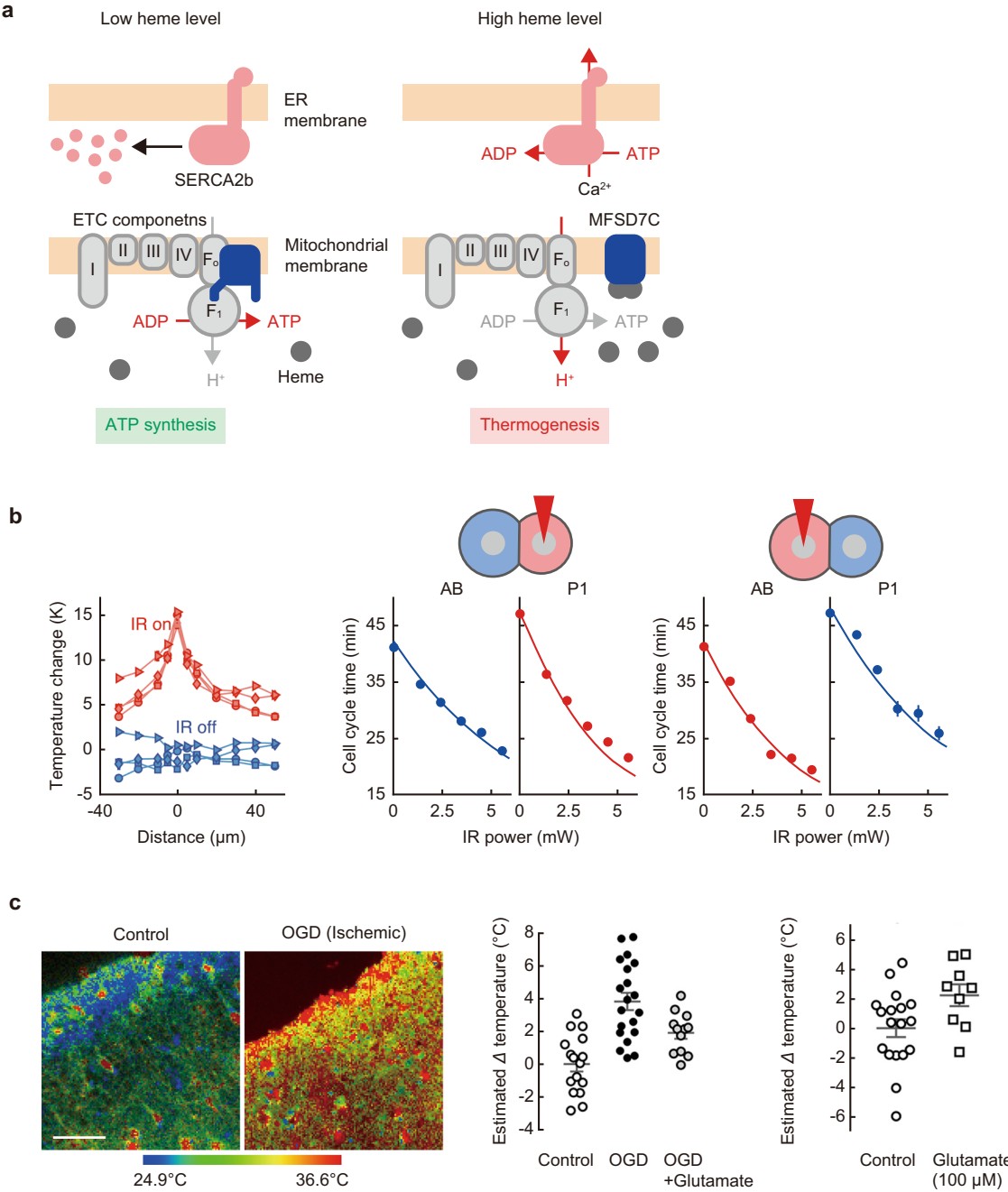

**Fig. 2 Cellular thermometry revealed intracellular temperature-dependent functions. a** Proposed model of MFSD7C functions as a switch between ATP synthesis (when the heme level is low) and thermogenesis (when the heme level is high) through its heme-dependent interaction with mitochondrial electron transport chain (ETC) components (I–IV). $F_OF_1$: ATP synthase, SERCA2b: sarcoendoplasmic reticulum $Ca^{2+}$-ATPase 2b. **b** Controlling the cell division timing in *C. elegans* embryos by local laser heating. Left, intracellular temperature change measured by a nanodiamond thermometer (in quadruplicate) while sweeping the position of an IR laser relative to the thermometer. Middle and right, selective acceleration of cell cycle time and its correlation with local temperature changes. Middle, P1 nucleus heating. Right, AB nucleus heating. Lines are theoretical predictions based on the average and nuclear temperatures of individual cells. The error bars denote the SDs. **c** Thermal signaling in mouse brain during ischemia. Left, representative images of temperature in nontreated (control) and oxygen-glucose deprivation (OGD)-treated brain slices. Scale bar, 50 μm. Middle and right, the estimated intracellular temperature change. This ischemia (OGD treatment)-derived increase in brain temperature via glutamate-dependent neuronal activity triggers TRPV4 channel opening. Panel **a** adapted from ref. [48], Springer Nature. Panel **b** adapted from ref. [50], National Academy of Sciences. Panel **c** adapted from ref. [51], Society for Neuroscience.

In the future, the careful pursuit of energy income and expenses in the thermal, chemical, and mechanical forms will lead to a deeper understanding of the scientific significance of intracellular temperature. Moreover, the construction of theories based on completely new concepts that can encompass the abovementioned related fields is also a challenging and promising direction for intracellular thermometry.

To understand intracellular temperature variations, it is necessary to experimentally verify and model three processes involved: thermogenesis, conversion to other energy types, and dissipation. For this purpose, a controllable heat source is useful because it allows quantifying the energy added to a cell, unlike heat production by chemical stimuli. Intracellular temperature mapping techniques with high temporal resolution will also contribute to kinetic studies. In addition, identifying biological macromolecules (i.e., nucleic acids, proteins, and lipids) involved in intracellular temperature regulation is another important step.

The use of intracellular thermometric technology is expected to advance research areas; 1. thermal medicine[61], which explores the manipulation of body or tissue temperature for the treatment of disease, including theragnostics[62], and 2. thermal biology, which consists of the measurement of intracellular temperature variations within individuals and the determination of temperature-sensing and temperature-maintaining mechanisms at the individual and single-cell levels. In particular, we are paying close attention to "thermal signaling" in the latter field, which is a signal transduction system utilizing temperature changes in the body as input, exemplified by the molecular switch that transitions cellular energy metabolism to heat production[48] and by TRPV4, which is activated by spontaneous temperature increases in its residing cell[51]. In contrast to the conventional viewpoint based on chemical signaling, "thermal signaling" is a new concept to define cellular functions by physical quantity in biology. Intracellular thermometry will greatly contribute to new biological studies to shed light on the significance of thermal signaling.

**Reporting summary**. Further information on research design is available in the Nature Research Reporting Summary linked to this article.

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

## Acknowledgements

We are grateful for financial support from PRESTO of JST and JSPS KAKENHI (17H03075 and 20H05785).

## Author contributions

S.U. organized this work. K.O. and S.U. wrote the paper with continuous discussion.

## Competing interests

The authors declare no competing interests.
