## [Peer Review File · Communications Biology]

Reviewers' comments:

Reviewer #1 (Remarks to the Author):

The authors provide a nice overview of recent work in the field of intracellular thermal biology, explaining the relevance of temperature variations and control to a wide range of biological phenomena, and focusing on a few results that depend specifically on intracellular temperatures. They argue potently about the relevance of intracellular temperatures and provide a nice perspective on the role of new measurement techniques and progress towards resolving some of the existing controversies in the field.

Overall, I find the perspective well-written, providing both a systematic overview of the field as well as new insights into the challenges and future directions of the field, and is accessible to readers from different backgrounds. Conditional on the following modifications, I believe that this is a suitable article for publication in *Communications Biology*.

The main shortcoming I see is that there is currently limited discussion on the measurement errors and artifacts in the current techniques, such as sensitivity to pH or other environmental factors. As an example, the nominal temperature distribution in Fig. 1b is quite wide, some values even going below zero Celsius, which makes direct interpretation of the mean value potentially more challenging. The consistency between very different measurement methods provides more confidence about the measurement artifacts, but a better understanding of the challenges and possible improvements associated with common measurement techniques, together with the advantages and disadvantages of different techniques, would still be a very good addition. There are also a few other techniques such as nanodiamond thermometry [e.g. Kucsko et al., *Nature* 500, 54] that might be worth mentioning. If there is limited space due to length requirements, I may even suggest reducing the length of the first section, which talks about the effect of macroscopic temperature differences and is less relevant to the microscopic scale.

Another direction that may be worth mentioning a few more words about, in relation to the heat shock proteins that have already been mentioned, is the active control and activation of cell functions via thermal control. An example of this that the authors may wish to add to citations include e.g. S. Hirsch et al., *Nat. Methods* 15, 921, and Y. Kamei et al., *Nat. Methods* 6, 79.

Regarding the discussion in Sec. 4, although the general discussion here seems reasonable, I have some questions about the arguments given in relation to internal consumption of energy. By itself, it seems to me that this would only decrease the temperature differences observable, rather than increase them? Naively, background diffusion still occurs and provides an upper bound on the amount of excess temperature, and any internal consumption will only further lower the temperature? The precise argument related to entropy should also be clarified further, given that in the absence of large heat diffusion outflows, one may imagine the system to be more like a closed system and total entropy should only increase.

Finally, although the authors do indeed focus in on thermal signaling at the very end of the article, most of the content, even about inhomogeneities and thermogenic activity, are only loosely related to this topic. In that sense, perhaps a different title that is broader and not just limited to thermal signaling may be more appropriate.

Reviewer #2 (Remarks to the Author):

The Perspective written by Okabe and Uchiyama provides a personal viewpoint on intracellular thermometry. The manuscript starts from introducing the importance of the temperature at large scales such as the body and the environment. Commenting on plants at the beginning should be a good approach when a broad readership of the current journal is considered. Then, the manuscript describes the temperature at the cellular scale, which is the main part of this manuscript. Recent progress in the field is introduced by citing relatively new references. The authors mainly cite their own studies and ignore many studies provided by other groups. However, it could be acceptable in

Perspective (but it should not be in Review). Yet, I request to cite two articles as commented below. To summarize, the manuscript is organized well with ideas that could stimulate discussion in the respective research area, and I find it potentially suitable for publication as Perspective in the current journal.

[Major comments]

(1) p4, L4 in "3. The benefits of intracellular thermometry"

The following statement is wrong: "The first methods for measuring intracellular temperature were developed by the combination of fluorescent molecular thermometers and optical microscopy in approximately 2010 by our group²³ and others²⁴." The authors should acknowledge at least two earlier studies in late 1990's.

(i) Chapman et al., *Photochem. Photobiol.* 62:416–425 (1995)

<https://doi.org/10.1111/j.1751-1097.1995.tb02362.x>

(ii) Zohar et al., *Biophys. J.* 74:82–89 (1998)

[https://doi.org/10.1016/S0006-3495\(98\)77769-0](https://doi.org/10.1016/S0006-3495(98)77769-0)

(2) Figure 2a and related description in page 3 in "3.2 Unveiled physiological significance of intracellular temperature variations"

Figure 2a and the corresponding texts could be misleading. The study by Li et al. cited as reference #40 in the current manuscript highlights the role of SERCA2b in MFSD7C-regulated cellular thermogenesis in their model. Their model describes not only the uncoupled mitochondrial respiration. Figure 2a in the current manuscript is reproduced well, but is oversimplified. To support an accurate understanding for readers, I suggest to add the illustration of SERCA2b in the current Figure 2a as in the original Figure 6 in reference #40, and corresponding changes in the legend.

Reviewer #3 (Remarks to the Author):

The work by Kohki Okabe and Seiichi Uchiyama provides a concise overview of the work performed during the past decade on intracellular thermometry and an interesting perspective on the major challenges of the field. The paper is well organized and very well written covering as a bird's-eye view the most important aspects of the field and is supported by an updated bibliography (with some missing important references, noted below). Thus, the manuscript has the potential to be an important reference on the topic and I undoubtedly recommend its publication in the journal. I have, however, a few suggestions to eventually ameliorate the paper, listed in what follows:

1. As several reviews were published in the past five years covering the advances of intracellular thermometry (e.g., 10.1002/smll.201600665, ref. 25, 10.1007/s12551-020-00683-8, 10.1039/C7CC06203F – not exhaustive) the last part of the introduction must be reformulated to mention those papers and state clearly what differentiates the present manuscript from those published works.
2. I suggest slightly enlarging the discussion about heterogeneous local thermogenesis (e.g., 10.1021/nn201142f).
3. On page 4, the references ascribing the pioneering work on the subject are not the most appropriate, note that ref. 24 is a review article from 2018.
4. The outlook is short and somehow limited lacking a perspective on the impact of intracellular thermometry in therapeutic technologies, for instance, in magnetic hyperthermia. This is, probably, the main weakness of this remarkable revision and I strongly suggest an additional effort from the authors to project the technological impact of precise intracellular temperature measurements.

[Point-by-point answers and revisions responding to reviewer 1's comments]

We appreciate your positive and constructive comments concerning our manuscript. Our point-by-point responses and revisions in accordance with reviewer 1's comments are below. We believe that the revised manuscript has been improved as expected by reviewer 1.

Comment: The authors provide a nice overview of recent work in the field of intracellular thermal biology, explaining the relevance of temperature variations and control to a wide range of biological phenomena, and focusing on a few results that depend specifically on intracellular temperatures. They argue potently about the relevance of intracellular temperatures and provide a nice perspective on the role of new measurement techniques and progress towards resolving some of the existing controversies in the field.

Response: This summary of our study is accurate.

Comment: Overall, I find the perspective well-written, providing both a systematic overview of the field as well as new insights into the challenges and future directions of the field, and is accessible to readers from different backgrounds. Conditional on the following modifications, I believe that this is a suitable article for publication in communications biology. The main shortcoming I see is that there is currently limited discussion on the measurement errors and artifacts in the current techniques, such as sensitivity to pH or other environmental factors. As an example, the nominal temperature distribution in Fig. 1b is quite wide, some values even going below zero Celsius, which makes direct interpretation of the mean value potentially more challenging. The consistency between very different measurement methods provides more confidence about the measurement artifacts, but a better understanding of the challenges and possible improvements associated with common measurement techniques, together with the advantages and disadvantages of different techniques, would still be a very good addition.

Response: Although we agree with the reviewer 1's opinion, a lot of comprehensive review articles (e.g., refs. 30, 31 and 33 in our case) summarizing the measurement errors and future possible improvements are now available elsewhere. To avoid substantial overlaps, we have just indicated important comprehensive review items with additional citations in the revised manuscript as follows:

"These achievements have accelerated the further establishment of intracellular temperature measurement methods using various fluorescent molecular thermometers or nonfluorescent metal materials represented by thermocouples; for more details about each method, their shortcoming and uncertainties, as well as potential improvements, readers can refer to the invaluable review articles²⁹⁻³⁵. Thus, we here focus on the contribution of intracellular thermometry to thermal biology, avoiding repetitive description on the comprehensive characteristics of the methods themselves." (Page 5, para 2, lines 7-12, the underlined part has been added in the revised manuscript)

"29. Bai, T., & Gu, N. Micro/nanoscale thermometry for cellular thermal sensing. *Small*, **12**, 4590-4610 (2016).

30. Uchiyama, S., Gota, C., Tsuji, T. & Inada, N. Intracellular temperature measurements with fluorescent polymeric thermometers. *Chem. Commun.*, **53**, 10976-10992 (2017).

32. Suzuki, M. & Plakhotnik, T. The challenge of intracellular temperature. *Biophys. Rev.*, **12**, 593-600 (2020)."

(Page 13 in the reference list in the revised manuscript, refs. 31 and 33-35 were cited in the original manuscript)

Comment: There are also a few other techniques such as nanodiamond thermometry [e.g. Kucsko et al., Nature 500, 54] that might be worth mentioning. If there is limited space due to length requirements, I may even suggest reducing the length of the first section, which talks about the effect of macroscopic temperature differences and is less relevant to the microscopic scale.

Response: Done as requested. According to the reviewer's suggestion, we have added the following description with the corresponding reference in the revised manuscript.

"Quantitative heating in *Caenorhabditis elegans* embryos calibrated by intracellular thermometry, with fluorescent nanodiamonds having relatively low chemical interference⁴⁹, promotes embryonic development in a fixed manner, indicating that embryogenesis is controlled by intracellular temperature (Figure 2b)⁵⁰." (Page 7, para 1, lines 3-6, the underlined part has been added in the revised manuscript)

"49. Kucsko, G. et al. Nanometre-scale thermometry in a living cell. *Nature*, **500**, 54-58 (2013)." (Page 15 in the reference list in the revised manuscript)

Comment: Another direction that may be worth mentioning a few more words about, in relation to the heat shock proteins that have already been mentioned, is the active control and activation of cell functions via thermal control. An example of this that the authors may wish to add to citations include e.g. S. Hirsch et al., *Nat. Methods* 15, 921, and Y. Kamei et al., *Nat. Methods* 6, 79.

Response: Done as requested. According to the reviewer's suggestion, we have added the following description with the corresponding references in the revised manuscript.

"Intracellular thermoresponsive molecules such as a heat shock promoter²³ and/or a temperature sensitive (ts) mutant protein²⁴ would allow for functional analyses of any given protein via spatiotemporally controlled heating within a cell." (Page 5, para 1, lines 7–9 in the revised manuscript)

"23. Kamei, Y. et al. Infrared laser-mediated gene induction in targeted single cells *in vivo*. *Nat. Methods*, **6**, 79–81 (2009).

24. Hirsch, S. M. et al. FLIRT: fast local infrared thermogenetics for subcellular control of protein function. *Nat. Methods*, **15**, 921–923 (2018)." (Page 13 in the reference list in the revised manuscript)

Comment: Regarding the discussion in Sec. 4, although the general discussion here seems reasonable, I have some questions about the arguments given in relation to internal consumption of energy. By itself, it seems to me that this would only decrease the temperature differences observable, rather than increase them? Naively, background diffusion still occurs and provides an upper bound on the amount of excess temperature, and any internal consumption will only further lower the temperature?

Response: The answers to the reviewer 1's questions are all yes. Nevertheless, this is not contradictory to our discussion in the manuscript due to the complexity of the biological cell. To imply it, we have added the following description in the revised manuscript.

"In actual biological cells, many endothermic reactions occur⁵⁵, and the energy stored in this way may be dissipated through subsequent exothermic reactions." (Page 7, last line–Page 8, line 2 the underlined part has been added in the revised manuscript)

Comment: The precise argument related to entropy should also be clarified further, given that in the absence of large heat diffusion outflows, one may imagine the system to be more like a closed system and total entropy should only increase.

Response: It is widely known that biological species (including a biological cell) exclusively maintain their entropy to be low by using energy. In other words, biological species cannot be considered closed systems. To make these clearer, we have cited an educational paper concerning the entropy in biological systems with the corresponding description in the revised manuscript as follows:

"Cells keep entropy low, a phenomenon found exclusively in living organisms⁵⁴." (Page 7, para 2, line 10 in the revised manuscript)

"54. Peterson, J. Evolution, entropy, & biological information. *Am. Biol. Teach.* **76**, 88–92 (2014)." (Page 15 in the reference list in the revised manuscript)

Comment: Finally, although the authors do indeed focus in on thermal signaling at the very end of the article, most of the content, even about inhomogeneities and thermogenic activity, are only loosely related to this topic. In that sense, perhaps a different title that is broader and not just limited to thermal signaling may be more appropriate.

Response: According to the reviewer's suggestion, the title of this *Perspective* has been changed into "**Intracellular thermometry uncovers spontaneous thermogenesis and associated thermal signaling**".

Thank you again for your kind comments.

[Point-by-point answers and revisions responding to reviewer 2's comments]

We appreciate your favourable and constructive comments concerning our manuscript. The following summarizes our point-by-point responses and revisions in accordance with the comments of reviewer 2. We believe that the revised manuscript will meet reviewer 2's expectations.

Comment: The Perspective written by Okabe and Uchiyama provides a personal viewpoint on intracellular thermometry. The manuscript starts from introducing the importance of the temperature at large scales such as the body and the environment. Commenting on plants at the beginning should be a good approach when a broad readership of the current journal is considered. Then, the manuscript describes the temperature at the cellular scale, which is the main part of this manuscript. Recent progress in the field is introduced by citing relatively new references. The authors mainly cite their own studies and ignore many studies provided by other groups. However, it could be acceptable in Perspective (but it should not be in Review). Yet, I request to cite two articles as commented below. To summarize, the manuscript is organized well with ideas that could stimulate discussion in the respective research area, and I find it potentially suitable for publication as Perspective in the current journal.

Response: This summary of our study is accurate.

Comment: [Major comments] (1) p4, L4 in "3. The benefits of intracellular thermometry"

The following statment is wrong: "The first methods for measuring intracellular temperature were developed by the combination of fluorescent molecular thermometers and optical microcopy in approximately 2010 by our group²³ and others²⁴." The authors should acknowledge at least two earlier studies in late 1990's.

(i) Chapman et al., *Photochem. Photobiol.* 62:416–425 (1995) <https://doi.org/10.1111/j.1751-1097.1995.tb02362.x>

(ii) Zohar et al., *Biophys. J.* 74:82–89 (1998) [https://doi.org/10.1016/S0006-3495\(98\)77769-0](https://doi.org/10.1016/S0006-3495(98)77769-0)

Response: Done as requested. According to the reviewer's suggestion, we have added the following description concerning early studies on cellular temperature measurements in the revised manuscript.

"The efforts to measure intracellular temperature began with the placement of fluorescently labeled lipids on cell membranes^{25,26}." (Page 5, para 2, lines 4–5 in the revised manuscript)

²⁵Chapman, C. F., Liu, Y., Sonek, G. J. & Tromberg, B. J. The use of exogenous fluorescent probes for temperature measurements in single living cells. *Photochem. Photobiol.*, **62**, 416–425 (1995).

²⁶Zohar, O. et al. Thermal imaging of receptor-activated heat production in single cells. *Biophys J.*, **74**, 82–89 (1998)." (Page 13 in the reference list in the revised manuscript)

Comment: (2) Figure 2a and related description in page 3 in "3.2 Unveiled physiological significance of intracellular temperature variations". Figure 2a and the corresponding texts could be misleading. The study by Li et al. cited as reference #40 in the current manuscript highlights the role of SERCA2b in MFSD7C-regulated cellular thermogenesis in their model. Their model describes not only the uncoupled mitochondrial respiration. Figure 2a in the current manuscript is reproduced well, but is oversimplified. To support an accurate understanding for readers, I suggest to add the illustration of SERCA2b in the current Figure 2a as in the original Figure 6 in reference #40, and corresponding changes in the legend.

Response: Done as requested. In the revised manuscript, Figure 2a and the corresponding caption have been changed as follows (continued to the next page):

"Figure 2. Cellular thermometry revealed intracellular temperature-dependent functions. a. Proposed model of MFSD7C functions as a switch between ATP synthesis (when the heme level is low) and thermogenesis (when the heme level is high) through its heme-dependent interaction with mitochondrial electron transport chain (ETC) components (I-IV). F₀F₁: ATP synthase, SERCA2b: sarcoendoplasmic reticulum Ca²⁺-ATPase 2b. (Page 10, Figure 2 caption; the underlined parts have been added in the revised manuscript)

Thank you again for your kind comments.

[Point-by-point responses and revisions in accordance with reviewer 3's comments]

We appreciate your favorable and constructive comments concerning our manuscript. Our point-by-point responses and revisions in accordance with reviewer 3's comments are below. We believe that the revised manuscript has been improved as expected by the reviewer 3.

Comment: The work by Kohki Okabe and Seiichi Uchiyama provides a concise overview of the work performed during the past decade on intracellular thermometry and an interesting perspective on the major challenges of the field. The paper is well organized and very well written covering as a bird's-eye view the most important aspects of the field and is supported by an updated bibliography (with some missing important references, noted below). Thus, the manuscript has the potential to be an important reference on the topic and I undoubtedly recommend its publication in the journal.

Response: We are happy that reviewer 3 valued the importance of our manuscript.

Comment: I have, however, a few suggestions to eventually ameliorate the paper, listed in what follows: 1. As several reviews were published in the past five years covering the advances of intracellular thermometry (e.g., 10.1002/sml.201600665, ref. 25, 10.1007/s12551-020-00683-8, 10.1039/C7CC06203F – not exhaustive) the last part of the introduction must be reformulated to mention those papers and state clearly what differentiates the present manuscript from those published works.

Response: Done as requested. According to the reviewer's suggestion, we have reformulated the last part of introduction with the indicated additional references in the revised manuscript as follows.

"These achievements have accelerated the further establishment of intracellular temperature measurement methods using various fluorescent molecular thermometers or nonfluorescent metal materials represented by thermocouples; for more details about each method, their shortcoming and uncertainties, as well as potential improvements, readers can refer to the invaluable review articles²⁹⁻³⁵. Thus, we here focus on the contribution of intracellular thermometry to thermal biology, avoiding repetitive description on the comprehensive characteristics of the methods themselves." (Page 5, para 2, lines 7–12, the underlined part has been added in the revised manuscript)

"29. Bai, T., & Gu, N. Micro/nanoscale thermometry for cellular thermal sensing. *Small*, **12**, 4590–4610 (2016).

30. Uchiyama, S., Gota, C., Tsuji, T. & Inada, N. Intracellular temperature measurements with fluorescent polymeric thermometers. *Chem. Commun.*, **53**, 10976–10992 (2017).

32. Suzuki, M. & Plakhotnik, T. The challenge of intracellular temperature. *Biophys. Rev.*, **12**, 593–600 (2020)." (Page 13 in the reference list in the revised manuscript, refs. 31 and 33–35 were cited in the original manuscript)

Comment: 2. I suggest slightly enlarging the discussion about heterogeneous local thermogenesis (e.g., 10.1021/nn201142f).

Response: Done as requested. According to the reviewer's suggestion, we have added the following description with the reference in the revised manuscript.

"For example, non-uniform temperature distribution due to spontaneous intracellular local heat generation was recorded²⁸." (Page 5, last line–Page 6, line 1 in the revised manuscript)

"28. Yang, J.-M. Yang, H., & Lin, L. Quantum dot nano thermometers reveal heterogeneous local thermogenesis in living cells. *ACS Nano*, **5**, 5067–5071 (2011)." (Page 13 in the reference list in the revised manuscript)

Comment: 3. On page 4, the references ascribing the pioneering work on the subject are not the most appropriate, note that ref. 24 is a review article from 2018.

Response: According to the reviewer's suggestion, we have replaced the original ref. 24 by the new reference 28 by Yang et al. (For the detail of ref. 28, see the above response on Comment 2)

Comment: 4. The outlook is short and somehow limited lacking a perspective on the impact of intracellular thermometry in theragnostic technologies, for instance, in magnetic hyperthermia. This is, probably, the main weakness of this remarkable revision and I strongly suggest an additional effort from the authors to project the technological impact of precise intracellular temperature measurements.

Response: Done as requested. According to the reviewer's suggestion, we have added the following description with corresponding references in the revised manuscript.

"The use of intracellular thermometric technology is expected to advance research areas: 1. thermal medicine⁶¹, which explores the manipulation of body or tissue temperature for the treatment of disease, including theragnostics⁶², and 2. thermal biology, which consists of the measurement of intracellular temperature variations within individuals and the determination of temperature-sensing and temperature-maintaining mechanisms at the individual and single-cell levels." (Page 9, para 1, lines 1–5, the underlined part has been added in the revised manuscript)

"61. Diederich, C. J. Thermal ablation and high-temperature thermal therapy: overview of technology and clinical implementation. *Int. J. Hyperth.* **21**, 745–753 (2005).

62. Wu, Y. et al. Nanodiamond theranostic for light-controlled intracellular heating and nanoscale temperature sensing. *Nano Lett.* **21**, 3780–3788 (2021)." (Page 16 in the reference list in the revised manuscript)

Thank you again for your kind comments.

REVIEWERS' COMMENTS:

Reviewer #1 (Remarks to the Author):

I am generally happy with the author's modifications, and most issues raised in the referee reports have been satisfactorily addressed. However, I would still prefer if the authors could further sharpen the arguments related to endothermic reactions and entropy reduction, and how that relates to filling in the "Baffou gap". In my mind, to fill in the gap, one needs a mechanism to increase heat production, rather than decrease it through endothermic reactions or entropy reduction. In other words, it seems to me that most of the discussion from "it does not accommodate any heat biologically consumed within a cell" to "were solely due to intracellular heat conduction", while all being correct statements, does not provide a concrete mechanism that can increase the temperature more than limits imposed by thermal conduction, and rather seems to imply that the temperature increase will be less. As a reader, the arguments about thermal conduction rates being different in cells seem more compelling. I believe that I am likely misunderstanding some of the arguments here, but in the interest of ensuring that readers do not make the same mistakes, it would be great if the authors could clarify things.

Reviewer #2 (Remarks to the Author):

My initial concerns have mostly been cleared. I only have one minor suggestion which relates to my previous comment (1).

The revised text now states that:

(page 5) "The efforts to measure intracellular temperature began with the placement of fluorescently labeled lipids on cell membranes^{25,26}"

To be more precise, Zohar et al. (Ref. #26) stained the cellular membranes with the rare earth chelate europium (III) thenoyltrifluoro-acetonate (Eu-TTA). In contrast to the method by Chapman et al. (Ref. #25), Zohar et al. did not conjugate the dye to lipids. Therefore, I suggest to say something like:

"The efforts to measure intracellular temperature began with the placement of fluorescently labeled lipids (Ref. #25) or hydrophobic thermosensitive dyes (Ref. #26) on cell membranes."

Reviewer #3 (Remarks to the Author):

The authors answered satisfactorily the referees' concerns and the paper is recommended for publication.

[Point-by-point answer and revision responding to reviewer 1's comments]

We appreciate your positive and constructive comments concerning our manuscript. Our point-by-point responses and revisions in accordance with reviewer 1's comments are below. We believe that the revised manuscript has been improved as expected by reviewer 1.

Comment: I am generally happy with the author's modifications, and most issues raised in the referee reports have been satisfactorily addressed. However, I would still prefer if the authors could further sharpen the arguments related to endothermic reactions and entropy reduction, and how that relates to filling in the "Baffou gap". In my mind, to fill in the gap, one needs a mechanism to increase heat production, rather than decrease it through endothermic reactions or entropy reduction. In other words, it seems to me that most of the discussion from "it does not accommodate any heat biologically consumed within a cell" to "were solely due to intracellular heat conduction", while all being correct statements, does not provide a concrete mechanism that can increase the temperature more than limits imposed by thermal conduction, and rather seems to imply that the temperature increase will be less. As a reader, the arguments about thermal conduction rates being different in cells seem more compelling. I believe that I am likely misunderstanding some of the arguments here, but in the interest of ensuring that readers do not make the same mistakes, it would be great if the authors could clarify things.

Response: As mentioned in the present manuscript (and in our old review articles [30,31]), we consider that to calculate an intracellular temperature variation by the Baffou's equation is scientifically inappropriate. This means that we need not fill the Baffou's gap derived from the inappropriate evaluation. Nevertheless, for the bright future in the interesting research field of intracellular thermometry, we prefer to accelerate discussion on the issue among researchers in this field in the present Perspective. Thus, we have added the following sentence in the revised manuscript:

"The dissipation of energy converted from heat is no longer heat conduction, and the consequence of this conversion on the temperature change in a cell, whether greater or lesser, is an open question." (Page 8, lines 5–7 in the revised manuscript)

Thank you again for your kind comment.

[Point-by-point answer and revision responding to reviewer 2's comments]

We appreciate your favourable and constructive comments concerning our manuscript. The following summarizes our point-by-point response and revision in accordance with the comments of reviewer 2. We believe that the revised manuscript will meet reviewer 2's expectations.

Comment: My initial concerns have mostly been cleared. I only have one minor suggestion which relates to my previous comment (1). The revised text now states that: (page 5) "The efforts to measure intracellular temperature began with the placement of fluorescently labeled lipids on cell membranes^{25,26}". To be more precise, Zohar et al. (Ref. #26) stained the cellular membranes with the rare earth chelate europium (III) thenoyltrifluoro-acetonate (Eu-TTA). In contrast to the method by Chapman et al. (Ref. #25), Zohar et al. did not conjugate the dye to lipids. Therefore, I suggest to say something like: "The efforts to measure intracellular temperature began with the placement of fluorescently labeled lipids (Ref. #25) or hydrophobic thermosensitive dyes (Ref. #26) on cell membranes."

Response: Done as suggested. According to the reviewer's suggestion, we have replaced the original sentence by the following one in the revised manuscript.

"The efforts to measure intracellular temperature began with the placement of fluorescently labeled lipids²⁵ or a hydrophobic thermosensitive dye²⁶ on cell membranes." (Page 5, para 2, lines 4–5 in the revised manuscript)

Thank you again for your kind comment.